# Detection of Putative Virulence Genes *alr, goiB*, and *goiC* in *Mycoplasma hominis* Isolates from Austrian Patients

**DOI:** 10.3390/ijms24097993

**Published:** 2023-04-28

**Authors:** Iwona Lesiak-Markowicz, Julia Walochnik, Angelika Stary, Ursula Fürnkranz

**Affiliations:** 1Institute of Specific Prophylaxis and Tropical Medicine, Centre for Pathophysiology, Infectiology and Immunology, Medical University of Vienna, 1090 Vienna, Austria; 2Outpatients Centre for the Diagnosis of Venero-Dermatological Diseases, Pilzambulatotrium Schlösselgasse, 1080 Vienna, Austria

**Keywords:** *Mycoplasma hominis*, *Ureaplasma* spp., virulence genes

## Abstract

In *Mycoplasma hominis*, two genes (*alr* and *goiB*) have been found to be associated with the invasion of the amniotic cavity, and a single gene (*goiC*) to be associated with intra-amniotic infections and a high risk of preterm birth. The syntopic presence of *Ureaplasma* spp. in the same patient has been shown to correlate with the absence of *goiC* in *M. hominis*. The aim of our study was to investigate the presence of *alr*, *goiB*, and *goiC* genes in two groups of *M. hominis* isolates collected from symptomatic and asymptomatic male and non-pregnant female patients attending an Outpatients Centre. Group A consisted of 26 isolates from patients with only *M. hominis* confirmed; group B consisted of 24 isolates from patients with *Ureaplasma* spp. as the only co-infection. We extracted DNA from all *M. hominis* isolates and analysed the samples for the presence of *alr*, *goiB*, and *goiC* in a qPCR assay. Additionally, we determined their cytotoxicity against HeLa cells. We confirmed the presence of the *alr* gene in 85% of group A isolates and in 100% of group B isolates; *goiB* was detected in 46% of the samples in both groups, whereas *goiC* was found in 73% of group A and 79% of group B isolates, respectively. It was shown that co-colonisation with *Ureaplasma* spp. in the same patient had no effect on the presence of *goiC* in the respective *M. hominis* isolate. We did not observe any cytotoxic effect of the investigated isolates on human cells, regardless of the presence or absence of the investigated genes.

## 1. Introduction

Mycoplasmas belong to the class Mollicutes, which are the smallest, self-replicating bacteria, lacking a rigid cell wall and possessing an extremely small genome [1]. These bacteria are mostly isolated from the mucosal surfaces of the genitourinary and respiratory tracts. *Mycoplasma hominis, Ureaplasma urealyticum*, and *Ureaplasma parvum* represent mycoplasmas associated with the normal genital flora, as well as with infection, and are very common [2,3]. These species do not appear to cause symptoms or adverse effects in non-pregnant women, and *M. hominis* seems to be associated with abnormal vaginal discharge only in women who also have bacterial vaginosis (BV) [3]. However, in pregnancy, colonisation with *M. hominis* or *Ureaplasma* spp. may be associated with several adverse pregnancy outcomes [4,5], and intraamniotic infections with these microorganisms have been associated with inflammation, preterm rupture of membranes, and preterm birth [6]. Allen-Daniels and colleagues identified two genes (*alr* and *goiB)* in *M. hominis* isolated from pregnant women to be associated with the invasion of the amniotic cavity and a single gene (*goiC*) to be associated with intra-amniotic infections and a high risk of preterm birth. The *alr* gene appears to encode an alanine racemase that converts L-alanine to D-alanine, which is a component of peptidoglycan. Although Mycoplasmas do not possess peptidoglycan, some species are able to encode alanine recamase [7]. However, there is no information available on the function of this enzyme in Mycoplasmas. The gene named by Allen-Daniels as *goiB* (gene of interest B) encodes a 379 amino acid protein of unknown function, which aligns over 97% of its length to a hypothetical protein from *U. urealyticum.* This gene is supposed to encode a secreted protease [7]. The *goi C* gene (designed by Allen-Daniels as gene of interest C) shows no similarity to proteins of known function, but appears, along with the previous two genes, to be unique to amniotic fluid/placental isolates. In that study, the absence of *goiC* in tested isolates was shown to correlate with the presence of *Ureaplasma* spp. in the vagina of the same patient [7].

The aim of the current study was to screen clinical *M. hominis* isolates from men and non-pregnant women for *alr*, *goiB*, and *goiC* and to evaluate if there is a correlation between the absence of *goiC* in tested isolates and the presence of *Ureaplasma* spp. in the same patient. Moreover, the cytotoxicity of the isolates against HeLa cells was investigated.

## 2. Results

### 2.1. Detection of the Putative Virulence Genes

The presence/absence of *alr*, *goiB*, and *goiC* in the respective isolates is listed in Table 1.

Twelve of the patients (six from group A and six from group B; four women and eight men) reported no symptoms; in all their *M. hominis* isolates, the presence of *goiC* was confirmed. The other 38 patients reported various symptoms, including pain, itching, vaginitis, and prostatitis.

### 2.2. Co-Occurrence of Putative Virulence Genes

The co-occurrences of the *alr*, *goiB*, and *goiC* genes in the two *M. hominis* groups are listed in Table 2.

### 2.3. Lactate Dehydrogenase Assay (LDH)

In the current study, none of the *M. hominis* isolates investigated, regardless of the presence or absence of the mentioned genes, showed toxicity to HeLa cells. After 48 h of incubation of *M. hominis* collected from LDH assay plates in PPLOa medium, a change in the colour of the PPLOa medium was observed in all samples collected, indicating that all *M. hominis* isolates survived and were capable of growing.

### 2.4. Growth Assay

There were no significant differences in growth rates between the study groups (A and B. 

### 2.5. Comparison of the Ct-Values

Comparison of the Ct-values of *hitA* and *goiC* within the same *M. hominis* sample revealed no differences. Both were constant between cycles 16 and 20, except for two *M. hominis* isolates, where *hitA* emerged at cycle 13 and 16 and *goiC* at cycles 23, and 19, respectively.

## 3. Discussion

We compared two groups of patients with *M. hominis* isolates and showed that the putative virulence genes (*alr*, *goiB*, and *goiC*) were present in both investigated groups. Co-colonisation with *Ureaplasma* spp. in the same patient had no effect on the presence of *goiC* in the corresponding *M. hominis* isolate. This is in contrast to the report that the absence of *goiC* in *M. hominis* isolates correlated with the co-existence of *Ureaplasma* spp. in the vagina [7]. We compared growth rates of the *M. hominis* isolates of groups A and B and revealed no growth impairment due to the presence or absence of any of the investigated genes, or the presence of *Ureaplasma* spp. in the same patient. The co-occurrence analysis of these genes also showed no significant differences between the tested groups.

In our collection of “*M. hominis* only” isolates (Group A), 85% of samples were positive for *alr*, 46% for *goiB*, and 73% for *goiC*. We compared our results with those available in public databases (n = 60) and found that only 16 *M. hominis* strains were positive for *alr* (26%), 5 for *goiB* (8%), and 11 for *goiC* (18%). Unfortunately, for these isolates, no information about the patients, their medical history, or co-infections was available. Moreover, we compared the specific sequences of the primers used in this study, as well as their products, with those available in public databases and could confirm that they are highly conserved, mainly *alr* with identities 100–99.35% and *goiB* with identities 100–99.42%, while *goiC* presented identities 100–97.40%. The significantly higher rates of putative virulence genes in our isolates can be explained by the likelihood of mixed infections with different *M. hominis* genotypes within the same patient (as all colonies of *M. hominis* on the agar plates were used), which cannot be completely excluded. On the other hand, a comparison of Ct values of the control gene *hitA* and the target gene *goiC* within the same sample of *M. hominis* revealed no difference, resulting in the assumption that both genes were present in the same number in each *M. hominis* isolate tested. Only in two of the 50 *M. hominis* isolates investigated, notably from the group where *Ureaplasma* spp. was also detected in the same patient, was there a discrepancy in the Ct values with regard to *goiC* emerging later than *hitA*. This could be an indication of genetic variation in these two isolates, or growth inhibition of *goiC*-positive *M. hominis* in the presence of *Ureaplasma* spp. Growth assays revealed no difference, regardless of the original presence or absence of *Ureaplasma* spp. in the same patient. Moreover, as this was only observed in two out of 24 isolates, where *M. hominis* and *Ureaplasma* spp. were detected, genetic variation seems more likely.

We can exclude the growth of other *Mycoplasma* species, because the agar plates on which *M. hominis* was cultured from swabs are designed to diagnose and distinguish between *M. hominis* and *Ureaplasma* spp. Isolates of *M. hominis* were checked for the presence of the putative virulence genes by PCR, always using *hitA* as a reference gene that is specific for *M. hominis*. Moreover, each patient was also tested for the presence of *M. genitalium* by specific PCR at the OCD, and only *M. hominis* isolates of *M. genitalium*-negative patients were included in the current study. In addition, testing more than 200 vaginal swabs collected at the OCD in our laboratory using the same 16S rRNA-specific primers revealed only *Candidatus* Mycoplasma girerdii, while no other mycoplasma species were detected except *M. hominis*, *M. genitalium*, *Ureaplasma parvum*, or *U. urealyticum* (own unpublished data).

Comparative studies on mycoplasmas have confirmed that each species possesses a unique or specific set of virulence genes, such as the lipoprotein genes in *M. penetrans* or *M. agalactiae* [8,9]. It is interesting to note that some of the major immunogenic lipoproteins (such as P35 in *M. penetrans*) can be completely switched off at the transcriptional level, making it difficult to detect these organisms in the state of disease. This means that infections caused by these microorganisms may go undetected [8]. The genes *alr*, *goiB*, and *goiC* are exceptional, as they cannot be switched on or off, but they are present or absent. However, it remains to be elucidated if these genes are indeed related to virulence, as in the current study, none of the *M. hominis* isolates investigated, regardless of the presence or absence of the mentioned genes, showed toxicity to HeLa cells. All isolates collected after co-incubation with HeLa cells were able to grow, and microscopic examination revealed no monolayer disruption of HeLa cells after co-incubation with the *M. hominis* isolates, indicating that there was no pathogenic effect on any of the *M. hominis* isolates. However, it was demonstrated that *M. hominis* mainly adhered to the cell surface of the HeLa cells, and mycoplasmal invasion was also observed. The ability of *M. hominis* to invade HeLa cells was first described by Taylor-Robinson [5]. We must point out that the ability of *M. hominis* to invade is not limited to HeLa cells, but has also been observed in spermatozoa [10,11] and *Trichomonas vaginalis*, which act as a ‘Trojan horse’ for *M. hominis* to replicate and distribute to other human hosts [12]. Thi Trung Thu et al. [13] demonstrated that a high number of *M. hominis* intracellularly associated with *T. vaginalis* possessed the *goiC* gene in association with *arl* and *goiB*. They also showed that metronidazole treatment of *M. hominis*-infected *T. vaginalis* allows the delivery of live intracellular *goiC*-positive *M. hominis* from antibiotic-killed protozoa and that these free *M. hominis* were able to infect human cell cultures, but no cytotoxicity assays were performed. Adherence and proteins involved in adherence are important for *Mycoplasma* pathogenicity, which is why most studies have focused on this [14,15,16,17]. Unfortunately, detailed studies on the cytotoxicity of *M. hominis* isolates carrying putative virulence genes are scarce, as are data on *alr* and *goiB* (which are also unique to *M. hominis* isolated from amniotic fluid or placenta) and their role in *M. hominis* virulence. In general, data on the role of *M. hominis* as a pathogen are rare and cannot be pinned down to single genes or proteins to date.

The participation of *Mycoplasma* spp. and *Ureaplasma* spp. in adverse pregnancy outcomes is increasingly accepted, and colonisation with both microorganisms at the same time has been linked with various adverse effects, including infertility, low birth weight, stillbirth, preterm delivery, pelvic inflammatory disease (PID), etc. [18,19]. Colonisation values of *M. hominis* on the mucosal surfaces of the cervix or vagina range between 20 and 30% around the world [18,19,20] and seem to indicate a commensal character of the microorganisms. Thus, the sole presence of these microorganisms in the vaginal flora may not cause pathological problems, but only their combination with other factors, such as bacterial vaginosis or cervical insufficiency, may induce preterm labour [21]. It has been shown that the colonisation of the lower genital tract with *M. hominis* and *U. urealyticum* was associated with placental tissue infection in women with spontaneous abortion. The authors assumed that the direct action of *U. parvum* on placental tissue altered the gestational tolerogenic state, reducing the immune response against pathogens and activating the extrinsic apoptotic pathway, causing spontaneous abortion [22]. As *M. hominis* and *Ureaplasma* spp. can be detected in both healthy women and BV patients, questions still arise as to whether *M. hominis* should be classified as a pathogen at all. To date, no evidence has been found that *M. hominis* is a vaginal pathogen in adults [23], and according to the recommendation of the European Academy of Dermatology and Venereology and the European STI Guidelines Editorial Board, patients should not be screened for *M. hominis*, *U. urealyticum*, and *U. parvum* [24]. However, it has been shown that the prevalence of *M. hominis* in semen samples of infertile and fertile men varies from 3.2% to 18.2% and from 0.9% to 14.3%, respectively [25]; the frequency of genital ureaplasmas and mycoplasmas detected in semen samples of infertile men in Tunisia was 19.2% and 15.8%, respectively [10]. Thus, the role of these microorganisms in disease is still not clear, and therefore, the question of whether *M. hominis* should be considered a pathogen still requires further research. Of the patients investigated in the current study, 12 did not report symptoms, despite all being carriers of *M. hominis* with *goiC*. This and the lack of cytotoxic activity towards HeLa cells in vitro suggests that perhaps *goiC* is not a virulence gene per se, or *M. hominis* with this gene is only virulent in pregnant women. Allen-Daniels and colleagues [7] focused on symptoms and isolation of *M. hominis* from the placenta to reflect the virulence of the isolates; no cytotoxicity assay was performed then. They described their results on the basis of just a few isolates. In our study also, no statistics were performed on the correlation of symptoms and the presence or absence of the genes tested, as a total of 50 samples is too small a number to allow reliable correlation statistics. Therefore, future studies involving a larger number of patients are needed to complete the picture. On the other hand, it might also be supposed that our negative lactate dehydrogenase assay results indicate that this test is not suitable for testing the virulence of *M. hominis* isolates, which has to be clarified in future studies. Furthermore, other cell lines, such as human placental cells or vaginal epithelial cells, should be tested in more detail to observe the possible cytotoxic effect of *M. hominis.* Also, testing of gene or protein expression levels during bacteria-human cell co-cultures could explain more about the role of *alr*, *goiC*, and *goiB* genes. However, in our study, we first focused on the presence or absence of these genes in two different *M. hominis* groups (with concomitant *Ureaplasma* spp. in the same patient and without). Exploration of the cytotoxicity of *M. hominis* isolates in other types of cells is planned for future projects.

Here, this question also arises: can *M. hominis* with putative virulence genes harm the foetus, as they can obviously enter amniotic fluid and pass the placenta? *M. hominis* has been isolated from various sites of the human body [1], including the central nervous system in neonates [1,26], indicating infection in utero or during birth. It still has to be elucidated if this extragenital *M. hominis* possess (more) virulence genes, and the question of whether to screen for *M. hominis* in pregnancy or not currently remains unanswered [27].

## 4. Materials and Methods

### 4.1. Sample Collection

Only patients for whom a complete STD screening (including tests for *Trichomonas vaginalis*, *Neisseria gonorrhoeae*, *Chlamydia trachomatis*, and *Mycoplasma genitalium* (Alinity m system (STI Amplification Kit; Abbott; North Chicago, IL, USA)), as well as *Candida* spp., aerobic and anaerobic bacteria including *M. hominis* and *Ureaplasma* spp. (by culture), or bacterial vaginosis (by Gram staining) was performed were considered for this study.

Genital swabs taken from symptomatic and asymptomatic patients visiting the Outpatients Centre for Diagnosis of Venero-Dermatological Diseases (OCD), Vienna, Austria, were smeared onto *Mycoplasma/Ureaplasma* agar plates (Thermo Fisher Scientific; Waltham, MA, USA) and cultivated anaerobically at 37 °C for two days before microscopic examination. *Mycoplasma genitalium* would not grow on these plates within the time of observation.

After microscopic examination, all colonies of a positive culture of *Mycoplasma hominis* (grown in typical fried egg shape) from each respective patient were further transferred and cultivated micro-aerobically in PPLOa, consisting of PPLO broth (Difco, Thermo Fisher Scientific) and supplemented with 10% arginine solution (Sigma-Aldrich, 100 mL/L), 20% yeast extract (Difco, Thermo Fisher Scientific, 50 mL/L), 0.2% phenol red indicator (VWR International GmbH; Radnor, Pennsylvania, USA), horse serum (Sigma-Aldrich; St. Louis, Missouri, USA, 100 mL/L), and an antibiotic mixture (Sigma-Aldrich, 10 mL/L); pH 6.5 at 37 °C. After two rounds of sub-culture at the ISPTM, the samples were used for LDH assay and PCR testing.

Altogether, only samples from patients with only *M. hominis* or *M. hominis* and *Ureaplasma* spp. were included in the current study. Out of those 50 isolates, 26 isolates were collected from patients with only *M. hominis* infection (11 from males and 15 from females), designated as group A, and 24 isolates were from patients with both *M. hominis* and *Ureaplasma* spp. as only co-infection (12 from males and 12 from females), designated as group B.

### 4.2. DNA Isolation

*Mycoplasma hominis* cultures in PPLOa were centrifuged at 10,000× *g* for 10 min, and after two washing steps with PBS (phosphate-buffered saline), the QIAGEN DNA Mini Kit (QIAGEN GmbH, Hilden, Germany) was used for the extraction of total DNA from the cultures. The DNA was stored at −20 °C until further use.

### 4.3. Detection of the Virulence Genes

Primers specific for *alr*, *goiB*, and *goiC* developed by Allen-Daniels [7] were used in a qPCR assay to detect these genes. The *M. hominis*-specific gene *hitA* was used as standard control [28].

qPCRs were performed in a CFX96 thermocycler (Bio-Rad; Hercules, CA, USA) in a total volume of 15 µL/sample, consisting of 7.5 µL Takyon No Rox SYBR Master Mix dTTP Blue (Eurogentec; Seraing, Liège, Belgium, USA ), 0.25 µL of each primer (10 µM), 5 µL water, and 2 µL DNA (10 ng) using following protocol: starting with 50 °C for 10 min and 95 °C for 10 min, followed by 35 repeats of 15 s at 95 °C and 1 min at 60 °C. The qPCR runs were conducted in duplicate, twice independently.

### 4.4. Lactate Dehydrogenase Assay

Determination of cytotoxicity was achieved with the Lactate Dehydrogenase Assay (CyQUANT^TM^ LDH Cytotoxicity Assay Kit reaction mixture, Thermo Fisher Scientific-Life Technologies), which is based on the release of lactate dehydrogenase (LDH) upon cell lysis. HeLa cells were used as target cells and cultured in 96-well plates (10*4 cells per well) in IMDM Medium (Gibco, Thermo Fischer Scientific), with 10% fetal bovine serum (Sigma Aldrich) at 5% CO_2_ for 24 h. For the test, the medium was replaced by 180 µL of IMDM medium without FBS, and 20 µL of *M. hominis* isolates (between 1 × 10^7^ cells/mL and 1 × 10^10^ cells/mL) were added to the HeLa cells and incubated for 2, 24, and 48 h at 37 °C in 5% CO_2_ atmosphere. Ten representatives from each study group (A and B) were chosen for the assays repeated three times, and mean cytotoxicity was calculated according to the manufacturer’s protocol. After distinct incubation times (2, 24, and 48 h), *M. hominis* samples were collected from the wells (total one well volume 200 µL) and transferred to PPLOa medium (content of each well in separate 2 mL medium) to evaluate survival; a change in the colour of the medium indicated the presence of viable *M. hominis*.

### 4.5. Growth Assay

Ten representatives of each study group (A and B) were cultivated in PPLOa medium to determine their growth, using colour-changing units (CCUs). Growth was determined after 4, 24, and 48 h of incubation, using serial 1:10 dilution in 96-well microtiter plates. Plates were sealed and incubated at 37 °C for 48 h. Each isolate was tested twice in duplicate. The number of bacteria was the highest dilution at which there was a visible colour change.

## 5. Conclusions

We presented that the presence of *Ureaplasma* spp. in the same patient had no effect on the presence of *goiC* in the respective *M. hominis* isolate. No cytotoxic effect of the tested isolates on human cells was found. Thus, the question of whether or not *M. hominis* should be considered a pathogen still requires further research.

## Figures and Tables

**Table 1 ijms-24-07993-t001:** Presence of *alr*, *goiB*, and *goiC* in tested *M. hominis* isolates groups; F—female patients and M—male patients.

Patients Isolates	Total Number F/M	*alr* (%) F(%)/M(%)	*goiB* (%) F(%)/M(%)	*goiC* (%) F(%)/M(%)
*M. hominis* only (group A)	26 15/11	22/26(85) 12(80)/10(90)	12/26(46) 6(40)/6(54)	19/26(73) 9 (60)/10(90)
*M. hominis + Ureaplasma* spp. (group B)	24 12/12	24/24(100) 12(100)/12(100)	11/24(46) 3(25)/8(67)	19/24(79) 10(83)/9(75)

**Table 2 ijms-24-07993-t002:** Co-occurrences of the putative virulence genes in tested *M. hominis* groups A and B.

Patients Isolates	*alr+/goiB+/goiC+* *(%)*	*alr+/goiB+/goiC-* *(%)*	*alr+/goiB-/goiC+* *(%)*	*alr+/goiB-/goiC-* *(%)*	*alr-/goiB-/goiC+* *(%)*	*alr-/goiB-/goiC-* *(%)*
*M. hominis* only (group A)	9/26 (34.6)	3/26 (11.5)	8/26 (30.8)	2/26 (7.7)	2/26 (7.7)	2/26 (7.7)
*M. hominis* *+ Ureaplasma* spp. (group B)	10/24 (41.7)	1/24 (4.2)	9/24 (37.5)	4/24 (16.6)	0/24 (0)	0/24 (0)

*alr+goiB+goiC+:* all genes detected. *alr+goiB+goiC-: alr* and *goiB* genes detected. *alr+goiB-goiC+: alr* and *goiC* genes detected. *alr+goiB-goiC-:* only *alr* gene detected. *alr-goiB-goiC+:* only *goiC* gene detected. *alr-goiB-goiC-:* no putative virulence gene detected.

## Data Availability

Not applicable.

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
