# Peer review of "Detection of Putative Virulence Genes alrgoiB, and goiC in Mycoplasma hominis Isolates from Austrian Patients"

_ijms, 2023, doi:10.3390/ijms24097993_

Round 1

Reviewer 1 Report

This well written article describes an informative study on the possible virulence of the presence of three specific genes in Mycoplasma hominis. More specifically, the absence of goiC in combination with the presence of Ureaplasma urealyticum bacteria was tested for correlation.Two groups of patients were defined for whom in all cases M. hominis could be cultured from vaginal samples. One group was defined as having only M. hominis positive in culture and the other group also had positive cultures for Ureaplasma urealyticum.

The outcome was that no differences in virulence on Hela cells were seen for M. hominis irrespective of the presence of specific genes.

These data are adding to the discussion on whether M. hominis, which is omnipresent in vaginal samples, can also be seen as a pathogen in pregnancy: probably not!

There are a few points that need to be answered to further improve the quality of this paper and make it suitable for publication.

Comments.

  1. The goal of this study is not very clear. Does the presence of specific genes, as indicated by a positive PCR, indicate that there is indeed virulence of a bacterium? Or is the presence of RNA, as indicator of the expression of these genes a better indicator of virulence? Please discuss.
  2. Please write ‘LDH’ (lactate dehydrogenase assay) more often in full, certainly the first time that it appears in the text, but also in the title in 2.3. and in paragraph 4.1. As it is now it is only in 4.4. written in full.
  3. In 2.1. is stated that 12 patients reported no symptoms. What was the reason that they visited the clinic?
  4. It appears that an internal control gene was used, ‘hitA’, for the PCR. This is not clearly stated in Methods. Please adjust.
  5. In 2.5. the Ct (not ’ct’) values are compared for two genes, ‘hitA’ and ‘goiC’ which are probably both single copy genes and should therefor be present at the same Ct value if the efficiency of both PCRs are comparable. A difference of Ct 16 and Ct 20, so 4 Ct means at least 10 fold difference in number of cells. So this is not very robust. One sample had even 10 Ct difference, this is huge. There thus seem to be technical difficulties with one or both PCRs. Please comment on this.
  6. Discussion, line 85: please add ’patients with’ after ‘two groups’. I suggest to start the next line with ‘Co-colonisation’ as a new sentence.
  7. Discussion, line 106: please replace ‘amount’ by ‘number’
  8. Discussion, lines 132- 136: please consider to mention these data in Results instead of in Discussion (or both).

In 4.4, line 232 and in 4.5. the sentences start with a number. Please avoid this and rewrite these sentences.

Reviewer 2 Report

Main remarks and comments for the authors of the reviewed article:

1)    In the Introduction it has been not explained the functional or biochemical activity of selected genes (alr, goiB, and goiC) and pathways related to the pathogenic activity. There is also no information about the protein encoded by these genes, the level of expression of these genes, or the presumed reason for their association with intra-amniotic infections.

2)    There are no statistical calculations related to obtaining correlation factors according to the aim of the study. A small number of people were surveyed for this type of study.

 3)      In the Discussion, the authors noticed that tested M. hominis isolates not showed a cytotoxic effect in LHD Assay with HeLa cells and any differences between isolates with- or without goiC (lines 173-176). This part of the study is not supported by sufficient research and correctly explained because only one cell type was used for cytotoxicity testing. HeLa cells are epithelial cells originating from adenocarcinoma and used alone are insufficient for that type of experiment.

For cytotoxicity evaluation of tested M. hominis isolates other types of cells should be additionally used, for example, Human Placental Cells or similar. To explain more about the role of tested genes of M. hominis in the context of virulence factors additional part of the study could be taken into account on the gene or protein expression level during in vitro with M. hominis and human cells co-cultures. 

4)    It is difficult to discuss the amount of the PCR products based on ct values (Discussion, 101-106 lines) if no information about the amount of DNA used for the PCR mixture. There is also no information about DNA concentration measurement after isolation and QC of DNA.

Minor editorial remarks

62 line, improve for capital letters in the word “table” used in all text of the manuscript;

66 line, remove the additional minus sign in the genotype description;

91 line, complete dot sign after “spp” abbreviation;

199 line, complete dot sign at the end of the sentence.

Reviewer 3 Report

The authors reported a study on the prevalence of the putative virulence genes alr, goi B and goiC in 50 M. hominis isolates from men and non-pregnant women and the correlation between goiC and the presence of Ureaplasma spp.. Results showed that the three genes were presented in 46% to 100% of the isolates in different groups, while no correlation of goiC with Ureaplasma spp. was detected. This study provided some further knowledge in understanding M. hominis infections in different patient populations. There are some commons below:

1.      “Materials and Methods”:

a.       4.2 DNA isolation, line 213. For how long did the M. hominis cultures were centrifuged?

b.      4.4. LDH Assay. Line 231, What’s the doses of M. hominis added to the HeLa cells? The ratio of the M. hominis to cell may affect the result.  Multiple doses may need to be tested for dose response. Line 235-236. What’s the volume was transferred to PPLOa medium?

2.      There is no statistical analysis for the comparisons of the prevalence of the genes in different groups. A column of p value should be added to Table 1 and Table 2. The conclusions need to be re-checked based on the statistical analysis result.
